# Endoscopic Resection for Gastric Subepithelial Tumor with Backup Laparoscopic Surgery: Description of a Single-Center Experience

**DOI:** 10.3390/jcm10194423

**Published:** 2021-09-27

**Authors:** Wei-Jung Chang, Lien-Cheng Tsao, Hsu-Heng Yen, Chia-Wei Yang, Joseph Lin, Kuo-Hua Lin

**Affiliations:** 1Department of General Surgery, Changhua Christian Hospital, Changhua 500, Taiwan; microvillis@gmail.com (W.-J.C.); yannick881009@gmail.com (L.-C.T.); joseph.lin27@gmail.com (J.L.); 2Division of Gastroenterology, Changhua Christian Hospital, Changhua 500, Taiwan; blaneyen@gmail.com (H.-H.Y.); 97601@cch.org.tw (C.-W.Y.); 3Artificial Intelligence Development Center, Changhua Christian Hospital, Changhua 500, Taiwan; 4College of Medicine, National Chung Hsing University, Taichung 400, Taiwan; 5Department of Electrical Engineering, Chung Yuan University, Taoyuan 320, Taiwan

**Keywords:** gastric subepithelial tumor, endoscopic resection, laparoscopic surgery, gastrointestinal stromal tumor, delayed perforation

## Abstract

The aim of this study was to analyze patients who underwent endoscopic resection (ER) for gastric subepithelial tumors (SETs) with a high probability of surgical intervention. Between January 2013 and January 2021, 83 patients underwent ER at the operation theater and 27 patients (32.5%) required backup surgery mainly due to incidental perforation or uncontrolled bleeding despite endoscopic repairing. The tumor was predominantly located in the upper-third stomach (81%) with a size ≤ 2 cm (69.9%) and deep to the muscularis propria (MP) layer (92.8%) but there were no significant differences between two groups except tumor exophytic growth as a risk factor in the surgery group (37% vs. 0%, *p* < 0.0001). Patients in the ER-only group had shorter durations of procedure times (60 min vs. 185 min, *p* < 0.0001) and lengths of stay (5 days vs. 7 days, *p* < 0.0001) but with a higher percentage of overall morbidity graded III (0% vs. 7.1%, *p* = 0.1571). After ER, five patients (6%) had delayed perforation and two (2.4%) required emergent laparoscopic surgery. Neither recurrence nor gastric stenosis was reported during long-term surveillance. Here, we provide a minimally invasive strategy of endoscopic resection with backup laparoscopic surgery for gastric SETs.

## 1. Introduction

With advances in upper endoscopy and its wide availability, gastric subepithelial tumors (SETs) are occasionally detected during health examinations or cancer screening tests. Gastric SETs are classified as neoplastic lesions that are either malignant or have malignant potential, including gastrointestinal stromal tumors (GISTs), carcinoid tumors, lymphomas, glomus tumors, and lymphangiomas and non-neoplastic lesions such as leiomyomas, schwannomas, inflammation, cysts, and ectopic pancreas [1,2]. The management of subepithelial lesions was tailored according to tumor characteristics, symptoms, comorbidities, and patient compliance with long-term surveillance. GISTs are the most common type of gastric SETs with origin in the muscularis propria (MP) layer [3], and surgical resection is recommended when the size is larger than 20–30 mm [1,4,5,6] because of their malignant potential. A combination of endoscopic and laparoscopic techniques was developed to achieve precise localization, minimal resection, and functional preservation [7] with the advantages of both. In Japan, laparoscopic and luminal endoscopic cooperative surgery (LECS), developed by a multidisciplinary team since 2008 [8,9,10,11], has been approved for national insurance coverage to resect gastric SETs.

The management of small (≤2 cm), asymptomatic gastric subepithelial tumors (SETs) remains inconclusive and the patients’ compliance for long-term follow-up was poor [12] since the necessity of regular endoscopic ultrasound (EUS) surveillance was recommended [1]. With improvements in endoscopic techniques and devices [13,14,15,16,17,18,19], endoscopic submucosal dissection (ESD) has been tailored to the management of gastric SETs according to tumor characteristics [20,21,22,23,24,25,26], not only for diagnostic accuracy but also for therapeutic resection. However, endoscopic resection for gastric SETs originating from the MP layer carries a relatively high risk of complicated perforation in R0 attempts for tumor resection. Perforation or bleeding is encountered at rates ranging from 0% to 15.6% and 0% to 8.2%, respectively, even with endoscopic hemostasis or when clips are used [17,20,21,22,27,28]. Cooperation between the endoscopist and the surgeon could provide a safe environment with backup laparoscopic surgery if endoscopic resection (ER) fails. In this study, we aimed to analyze patients who underwent endoscopic resection for gastric SETs with a high probability of surgical intervention and to outline a feasible procedure with backup surgery.

## 2. Materials and Methods

### 2.1. Study Population

In this retrospective study, we examined the electronic medical records of 86 patients (three patients were later excluded due to anatomic changes) who underwent endoscopic resection for gastric SETs at the operation theater from a single medical center in Taiwan between January 2013 and January 2021. The study was approved by the Institutional Review Board of Changhua Christian Hospital (Document no. 201022 and no. 210202), and this retrospective study waived the need for informed consent.

Complete resection of gastric subepithelial tumors (SETs) is indicated for tumors >2 cm that are symptomatic with malignant features that are increasing in size during surveillance, or with patient’s preference [1,2,4,5,6]. For patients who declined either periodical surveillance of their SETs or laparoscopic resection, they received endoscopic resection in the operation theater with the backup laparoscopic surgery due to a high risk of incidental gastric perforation or uncontrolled bleeding. The high probability of surgical interventions included the following situations: (1) tumor located in the upper-third stomach due to a relatively thin wall and difficulty of endoscopic angulation; (2) tumor located at the anterior wall of the body, where air leakage easily occurred with difficult endoscopic repair; (3) tumor located deep to the MP layer or subserosa; and (4) tumor growth with an exophytic pattern. In total, 83 patients who underwent endoscopic intervention in the operation theater were included. Three patients with anatomic changes in the stomach due to previous operations were excluded, with two patients receiving subtotal gastrectomy and one who underwent esophagectomy.

### 2.2. Procedure

All patients received general anesthesia and prophylactic antibiotic (cefazolin 1 g) by intravenous infusion in the supine position at the operation theater. The gastroenterologists inserted an endoscope (GIF-H260Z or GIF-2TQ260M, Olympus, Tokyo, Japan) into the stomach of the patient and identified the tumor location. Two types of endoscopes were chosen depending on the location of the gastric SET: GIF-2TQ260M for those in the cardia and GIF-H260Z for those in the body or antrum. Endoscopic resection was performed around the tumor with a 0.5-cm margin by submucosal injection with glycerin, mucosal incision, and submucosal dissection. The DualKnife (KD-650L, Olympus, Tokyo, Japan) with an additional ITknife nano electrosurgical knife (KD-612L, Olympus, Tokyo, Japan) or Coagrasper Hemostatic Forceps (FD-412LR, Olympus, Tokyo, Japan) was used for endoscopy, depending on the patients’ preferences and budgets. The endoscopists attempted to remove the gastric SETs with R0 resection by preserving the tumor capsule like GIST or leiomyoma. Occasionally, piecemeal resection technique aimed to achieve R1 resection if R0 resection was not feasible in the situations of uncontrolled bleeding, large tumor size, or infiltrative tumor border with its surrounding gastric tissue, like aberrant pancreas or lipoma. In cases involving incidental gastric perforation or uncontrolled bleeding that could not be resolved by endoscopic clips or hemostasis, general surgeons took over the procedure and completed the procedure with laparoscopic wedge gastrectomy.

The patient was placed in the reverse Trendelenburg position. The general surgeon stood on the patient’s right side with an assistant on the contralateral side. A camera port was inserted into the umbilicus using the Hasson open technique to create a pneumoperitoneum with a pressure of 10–12 mmHg. Two additional ports (one 12-mm port and one 5-mm port) were inserted into the midclavicular line of the left-upper and right-upper abdomen with slight modifications depending on the lesion site. Tumors were easily identified due to incidental perforation by endoscopic resection (Figure 1). If the tumor was on the posterior stomach, the gastrocolic ligament was divided using a vessel-sealing device (LigaSure, LF1837, Covidien, Minneapolis, MN, USA) to flip the stomach and localize the lesions. The surgeon completed tumor resection with a 1-cm margin and closed the perforation site longitudinally to the stomach using a laparoscopic stapling device (Echelon Flex Endopath SC60A, Ethicon, USA) (Figure 2). The laparoscopic stapling device was adjusted in the transverse direction of the stomach if the tumor was located on the lesser curvature side or cardia to avoid stenosis. The staple line was further reinforced by continuing Lambert suturing with an absorbable wound closure device, 3-0 V-Loc (VLOCL0604, Covidien, USA) (Figure 3). The camera port was changed to the left-upper port in cases where it was difficult to locate the tumor or in cases of limited adjustment direction for stapling devices. The tumor was retrieved (Figure 4) using a specimen bag, and the absence of bleeding was confirmed before a drainage tube was placed near the perforation site or in the left subphrenic area.

### 2.3. Clinicopathological Factors

The clinical characteristics of the patients included age, sex, procedure time, length of hospital stay, and overall morbidity. All complications within 30 days utilized the revised Clavien–Dindo classification, with grades III–V considered clinically relevant morbidity [29]. The clinicopathological characteristics of the tumors were analyzed based on location, size, layer of tumor depth, pathological categories of malignant or malignant potential, and benign diseases. The location of the gastric SETs was classified by dividing the stomach into three equal sections (upper, middle, and lower third) or four equal sections (anterior wall, posterior wall, lesser curvature, and greater curvature). The diameter of the tumor was classified as ≤2 cm or >2 cm. Endoscopic ultrasonography was performed to determine the layer of tumor depth from the submucosal (SM) or MP layer. All patients received postoperative care with intravenous fluid supplement, pain relief, and intravenous esomeprazole (40 mg/day). Patients were allowed to attempt an oral diet on postoperative day (POD) 3 with an increasing amount as tolerated. Patients were discharged if they tolerated oral medication. Delayed perforation was identified after completion of ER with the sudden onset of symptoms related to pneumoperitoneum, as confirmed by an image survey. A further endoscopy was arranged 3 months after the surgery as a follow-up.

All specimens were analyzed by histological examination at the Department of Pathology at Changhua Christian Hospital. Immunohistochemistry analysis was performed with common markers such as CD117 (c-kit), CD34, DOG-1, smooth muscle actin, and S100. The c-kit protein is highly sensitive and specific for GISTs, whereas CD34 and DOG-1 are expressed in approximately 80% of GISTs. Positive staining for smooth muscle actin indicates the presence of leiomyoma or glomus tumor, while the presence of S100 indicates neural origin or schwannoma. GISTs are further divided by risk according to the National Institutes of Health-Fletcher classification after evaluation of the size and mitotic index by hematoxylin and eosin staining.

### 2.4. Statistical Analysis

Patient data were expressed as median with interquartile range (IQR) and categorical variables with percentage. A Mann–Whitney U-test was used to compare the differences in continuous variables, while a Chi-squared test was used to compare the differences in categorical variables. Statistical significance was defined as *p* < 0.05. All statistical analyses were performed on a personal computer using MedCalc for Windows version 20 (MedCalc Software, Ostend, Belgium).

## 3. Results

A total of 83 patients were enrolled for the study, of which 56 (67.5%) underwent ER-only for gastric SETs at the operation theater, while 27 (32.5%) underwent backup laparoscopic surgery. Among those receiving backup surgery, 26 cases were indicated due to incidental perforation (31.3%), with four cases of additional bleeding (4.8%) and one case was due to ill-defined tumor margins (1.2%). Overall, the tumor was predominantly located in the upper-third stomach (81%) with a size ≤ 2 cm (69.9%) and deep to the MP layer (92.8%). None of the factors, including age, sex, tumor location, tumor size, layer of tumor depth, whether pathologically proven malignancy or malignancy potential, and benign lesions, was statistically associated with the two resection methods. Although a higher percentage of tumor size ≤ 2 cm in the ER-only group (75% vs. 59.3%, *p* = 0.1455) and a higher percentage of tumor depth to the MP layer in the surgery group (100% vs. 89.3%, *p* = 0.0792) were observed, there did not exist a statistical difference. The ER-only group had shorter duration of procedure times (60 min vs. 185 min, *p* < 0.0001) and length of stay (5 days vs. 7 days, *p* < 0.0001) but also a higher percentage of overall morbidity, with grade III occurring, which showed no significant difference (7.1% vs. 0%, *p* = 0.1571) In the surgery group, a higher percentage of tumors with exophytic growth was noted (37% vs. 0%, *p* < 0.0001) (Table 1).

Among those patients with complications, two patients were graded IIIa, with one case of gastric ulcer bleeding receiving endoscopic hemostasis on POD 9 and another case of delayed perforation. Another two patients with grade IIIb delayed perforation underwent emergent laparoscopic surgery. Both cases were discharged on postoperative day 6 with uneventful recovery. No mortality was recorded during the follow-up period. Overall, five cases of delayed perforation (6%) are shown in Table 2. Among them, four patients had tumors located in the upper-third stomach and three patients had tumors in the anterior wall. All patients received endoclipping for suspected endoscopic perforation. Among them, two patients recovered from conservative treatment (2.4%), one case required radiological intervention (1.2%), and two patients underwent emergent laparoscopic surgery (2.4%).

For pathologic diagnosis, there were no significant differences in the malignancy or malignancy potential and benign lesions between the two groups. The detailed pathology is listed individually (Table 3). The most common type of malignant group was GIST (47%), followed by neuroendocrine tumor (1.2%), and hyperplastic polyps with focal high-grade dysplasia (1.2%). All tumors achieved R0 resection with a safe margin. According to the NIH classification [30], the recurrence risk of GISTs was categorized into very-low-risk (56.4%), low-risk (30.8%), and intermediate-risk (12.8%) groups. Patients diagnosed with GIST underwent long-term surveillance with a mean duration of 19.5 months, and no recurrence was detected. In the case of a neuroendocrine tumor Grade 1, the patient received further gastrectomy and adjuvant chemotherapy. No recurrence was noted during a 52-month follow-up period.

The most common benign group was leiomyoma (37.4%) followed by ectopic pancreas (3.6%), calcifying fibrous tumor (2.4%), lipoma (1.2%), inflammatory fibroid polyp (1.2%), gastritis cystica profunda (1.2%), plexiform fibromyxoma (1.2%), elastofibroma (1.2%), and pyloric gland adenoma (1.2%).

The location of the tumors and the percentage of patients requiring surgery, including two cases of delayed perforation receiving surgery, are depicted in Figure 5. Most tumors were in the upper-third stomach (*n* = 68, 81.9%), comprising the posterior wall (*n* = 27, 32.5%), cardia (*n* = 19, 22.9%), anterior wall (*n* = 15, 18.1%), and greater curvature (n = 7, 8.4%) in descending order. Fewer tumors were in the lower-third (*n* = 11, 13.3%) and middle-third stomach (*n* = 4, 4.8%). The overall surgery rate was 34.9%, and in descending order were upper (36.8%), middle (25%), and lower (27.3%). The highest surgery rate was noted in the anterior wall of the lower-third stomach (60%) followed by the lesser curvature of the middle-third (50%) and posterior wall of the upper-third (48.1%).

## 4. Discussion

Among 83 patients who underwent endoscopic resection for gastric SETs at the operation theater, 27 patients (32.5%) underwent backup laparoscopic surgery, and the most common indication was incidental perforation (96.3%). Our study demonstrated that gastric SETs highly selected for surgical intervention were mainly located in the upper-third stomach (81%) with size ≤ 2 cm (69.9%) and deep to the MP layer (92.8%). The risk factor for backup surgery was tumors with exophytic growth. Although there were limited cases in this study, tumors at the anterior wall and at the lesser curvature side were also considered. The ER-only group had shorter duration of procedure times (60 min vs. 185 min, *p* < 0.0001) and length of stay (5 days vs. 7 days, *p* < 0.0001) but also a higher percentage of overall morbidity with grade III occurred (7.1% vs. 0%, *p* = 0.1571). Five cases (6%) developed delayed perforation, and two patients (2.4%) underwent emergent laparoscopic surgery with uneventful recovery.

In our study, we modified the LECS procedure and adopted a novel strategy of a minimally invasive procedure that these patients received for endoscopic resection for gastric SETs under general anesthesia with backup laparoscopic surgery in the operation theater. For these selected patients, they declined either periodical surveillance of their SETs or laparoscopic resection and preferred this minimally invasive approach. Additionally, endoscopic resection is deemed to require general anesthesia to provide a more comfortable and safer environment because the endoscopist might have difficulty in approaching and managing the SETs considering endoscopic angulation and patient’s uncooperative status. Maintaining airway patency and avoiding aspiration pneumonia in the case of upper GI bleeding during endoscopic resection is another concern for safety. Once the endoscopist could not handle the incidental gastric perforation even after endoscopic repairing, the surgeon could take over the procedure without transportation between the endoscopy room and the operation room and avoid aggravating the risk of hemorrhagic shock or sepsis. Although LECS carries a risk of gastric content spillage and peritoneal seeding of tumor cells due to an intentional gastric perforation, the endoscopists attempted to remove the gastric SETs with R0 resection and the laparoscopists could efficiently clean up the perforation site without further contaminated fluid spreading. Most tumors were retrieved perorally. Neither intra-abdomen abscess postoperatively nor peritoneal seeding of tumor cell were reported during long-term surveillance. Thus, we believed some highly selected patients could benefit from this procedure.

With the concept and advances in LECS [7,8,9,10,11] by a multidisciplinary team, we modified and simplified the backup surgery after endoscopic resection [31] regardless of tumor location. Although a previous retrospective study demonstrated that laparoscopic wedge resection could also treat tumors in an unfavorable location in the lesser curvature or the posterior wall of the gastric body, fundus, and antrum [32], different methods with endoscopic submucosal tunneling [18], laparoscopic transgastric approach [32,33], and anatomic gastrectomy [34] have been reported to avoid lumen stenosis in tumors located near the gastroesophageal junction or the prepyloric area. In our study, once the tumor was easily identified after incomplete ER, the surgeon completed resection using a laparoscopic stapling device and adjusted to the transverse direction of the stomach in the junction area to minimize the resected volume and preserve the greatest function. Most patients were able to attempt an oral diet on postoperative day 3, and none of them had gastroesophageal junction stenosis or gastric outlet obstruction during long-term surveillance. This minimally invasive method is safe and feasible for the management of tumors at unfavorable locations without requiring anatomic gastrectomy. Furthermore, 27 patients received backup surgery in our study with the mean (standard deviation [SD]) 194.5 (67.8) min of operation time and the mean (SD) 7.3 (1.2) days of postoperative hospital stay, and no patients occurred complications above Clavien–Dindo grade III. Compared to a retrospective multicenter study from Japan [35], 126 patients received LECS for gastric SMT between October 2007 and December 2011 with the mean (SD) 190.2 (66.8) min of operation time and the mean (SD) 9.8 (10.1) days of postoperative hospital stay. Two patients (1.6%) had major morbidities with Clavien–Dindo grade IIIb/IVa due to leakage. Although a relatively small sample size in our study, surgical outcomes of those patients receiving backup surgery was comparable to the previous study with satisfactory outcomes.

In our study, the incidental perforation rate after endoscopic resection was 31.3%. In a review of 18 articles, the perforation rate was 3.02% after ESD and 98.8% of patients recovered under different methods of endoscopic repair without surgical intervention [36]. In two retrospective studies of ESD for gastric SETs, risk factors for incomplete resection included tumors in the upper-third stomach, tumors > 2 cm, and tumors originating from the MP layer [23,24]. The review study, which encompassed the field of early gastric cancer, further described other risk factors of intraoperative perforation, including location in the middle third, greater curvature and remnant stomach, tumor size, invasion depth, and submucosal fibrosis [36]. In our study with similar results, 81.9% of tumors were located in the upper-third stomach with a slightly higher percentage of 85.2% in the surgery group; 69.9% of tumors had a size of no more than 2 cm with a higher percentage of 75% in the ESD-only group; and 92.8% of tumors were located deep to the MP layer with a high percentage of 100% in the surgery group. No statistical difference was noted, probably because these cases were highly selected by the endoscopists in this setting. For tumors located in the upper-third stomach, the overall surgery rate was 36.8%, comprising a backup surgery rate of 33.8% and a delayed surgery rate of 3%. All tumors with exophytic growth necessitated a backup surgery. Thus, we would recommend laparoscopic wedge surgery for tumors with exophytic growth patterns and concern surgery backup for ER, while the tumor is located at the upper-third stomach and deep to the MP layer.

Our study reported five cases (6%) of delayed perforation and two patients (2.4%) requiring emergent laparoscopic surgery. From the review study, a low incidence rate of delayed perforation was reported to be 0.04–0.7% after gastric endoscopic resection [36], and most patients could be treated successfully with conservative care. Rare cases required emergent surgery at a rate of 0.043–0.45% within 1–2 days, with the most common indication being peritonitis [37,38,39,40,41]. Operation methods included simple closure, omentoplasty, and gastrectomy with hospital stays that ranged from 12–33 days [37,41]. In our study, all patients with delayed perforation had endoclipping during ER and developed symptoms within 2 days after ER, with a hospital stay of 8–9 days. Two patients underwent emergent laparoscopic gastrectomy with indications of internal bleeding and peritonitis, respectively. Previous studies reported that the risk factors of delayed perforation include tumor location in the upper-third stomach [41,42], prolonged procedure time [39,41,43], and exposure of the MP layer [43]. In our study, four patients had tumors located in the upper-third stomach and three patients had tumor depth at the MP layer, but all had a procedure time of less than 2 h. The mechanism of delayed perforation might be explained by ischemia changes [37,39,40,42] and transmural air leak [43]. The upper-third stomach, with a relatively thin wall and larger diameter of the submucosal arteries, was vulnerable to extensive submucosal dissection and repeated hemostasis by electrocautery, which might cause ischemia change and gastric wall necrosis. Frequent bleeding and impaired endoscopic view prolonged the procedure time, resulting in increased intragastric pressure, reduced intramural blood flow, and aggravated ischemia change. In addition, exposure or slight damage of the MP with potential transmural burn during ESD could result in intra-abdominal transmural air leaks. Once the perforation hole is embedded into the surrounding fat tissue, hemostasis with thermal injury to the peripheral vessels might cause delayed bleeding. Although two experienced endoscopists in our procedure minimized the risk of procedure time, other risk factors including tumor location in the upper-third stomach, tumor depth to the MP layer, and suspected perforation with endoclipping used during ER reminded us of timely recognition and effective management for delayed perforation in the previous two days after ER.

GISTs, accounting for 47% of all resected gastric SETs in this study, are the primary targets of treatment. However, GISTs are often difficult to differentiate from non-neoplastic lesions based on endoscopic sonography. Laparoscopic surgical resection is recommended for GISTs less than 5 cm and in a favorable location [5,32,44], while recent studies have reported that endoscopic resection is also feasible in a less invasive way [45,46,47]. Considering the malignant potential of GISTs [48,49], the US National Comprehensive Cancer Network and Japanese guidelines recommend either endoscopic or laparoscopic resection for tumors <2 cm if they are symptomatic or have high-risk features with rapid growth, ulceration, irregular margins, or heterogeneous echo patterns [5,6]. Recent retrospective studies found that 1.4 cm was an appropriate cutoff tumor size for small GISTs due to their potential for rapid tumor progression [50]. Endoscopic resection for small GISTs <2 cm is also safe and effective, which could confirm the diagnosis, improve the symptoms, reduce psychological pressure, and achieve complete cure [51]. Thus, some patients in our study preferred resection over long-term periodic surveillance. For GIST originating from the MP layer, receiving ER with backup surgery was safe and effective for patients in our setting. Long-term surveillance with a mean duration of 19.5 months was performed, and no recurrence was detected.

Our study has some limitations. First, the study was conducted retrospectively at a single center. Second, a relatively small sample size was analyzed because we focused on the patients who declined periodical surveillance of their SETs and laparoscopic resection. Although these studied populations compromised the minority of the patients with gastric SET, the current study is valuable in providing more information regarding the outcomes after endoscopic resection with backup laparoscopic surgery for patients who prefer a less invasive approach for tumor resection. Third, patients who underwent ER for tumors in the superficial submucosal layer (such as neuroendocrine tumor) in the endoscopic room were not included in this database. Selection bias resulted in the majority of tumors being located in the upper-third stomach and origin from the MP layer, which were considered risk factors for incidental perforation after ER. The rate of incidental perforation after ER requiring backup surgery and delayed perforation might be overestimated in this setting compared to other studies. Although there were limited cases in the other sites, our procedure could be applied to wherever the tumor was located in the stomach. Fourth, this study focused on gastric SETs, and SETs in the duodenum or esophagus were not investigated in the present study. Fifth, with the development of ER techniques, endoscopic submucosal tunneling was applied to eight cases with tumors located in the cardiac area, and none of them required further surgery. Procedure-related factors concerning different methods of ER techniques were not discussed in this study and wait for further analysis. Sixth, we lacked the data of those patients who received laparoscopic surgery for gastric SETs in our institution. Further research is required to investigate the comparison between the group of ER with backup surgery and the group of laparoscopic surgery [52,53].

## 5. Conclusions

Based on our center’s experience, gastric SETs with a high probability of surgical intervention are mainly located at the upper-third stomach with size ≤ 2 cm and deep to the MP layer with a backup surgery rate of 32.5%. As a backup surgery after ER, laparoscopic wedge gastrectomy can be applied to gastric SETs wherever they are located. The ER-only group had shorter duration of procedure times (60 min vs. 185 min, *p* < 0.0001) and lengths of stay (5 days vs. 7 days, *p* < 0.0001) but a higher percentage of overall morbidity with grade III (0% vs. 7.1%, *p* = 0.1571) occurring. Five of 56 patients had delayed perforation within 2 days and two required emergent laparoscopic surgery. Tumor with exophytic growth was the risk factor for backup surgery. Neither recurrence nor gastric stenosis was reported during long-term surveillance. Backup laparoscopic wedge gastrectomy is feasible and effective after incomplete ER for gastric SETs.

## Figures and Tables

**Figure 1 jcm-10-04423-f001:**
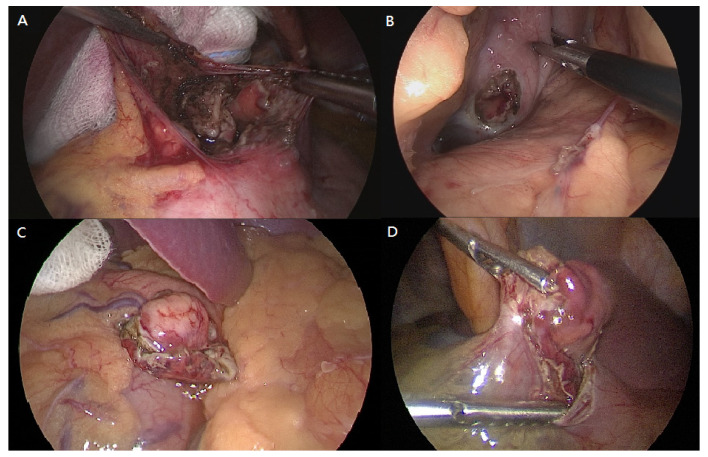
Different perforation sites of the stomach after endoscopic resection: (**A**) at the cardia, (**B**) at the posterior fundus, (**C**) at the greater curvature of fundus with incomplete resection, (**D**) at the anterior antrum with incomplete resection.

**Figure 2 jcm-10-04423-f002:**
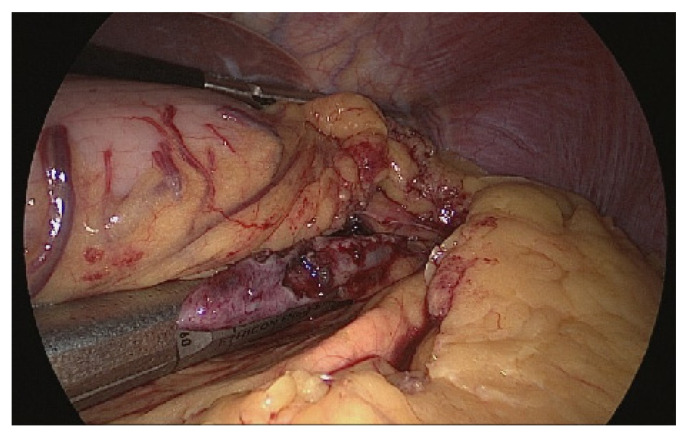
Laparoscopic stapling device used for closure of the gastric perforation site at the posterior fundus.

**Figure 3 jcm-10-04423-f003:**
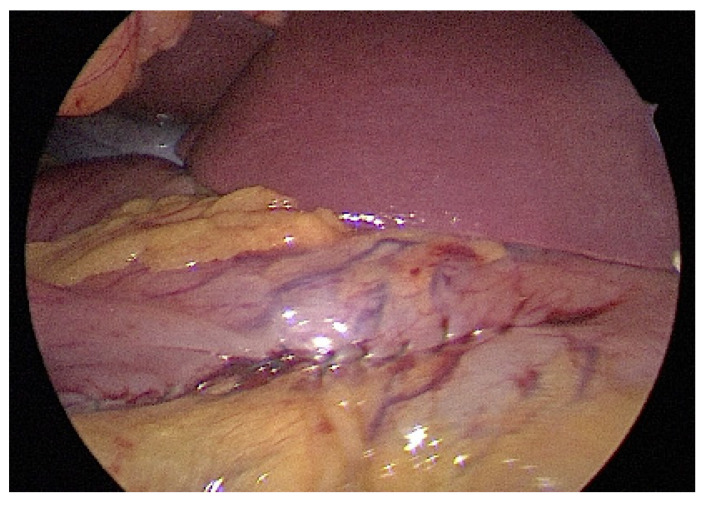
Laparoscopic reinforced suturing over the stapling line after closure of the perforation site at the anterior antrum.

**Figure 4 jcm-10-04423-f004:**
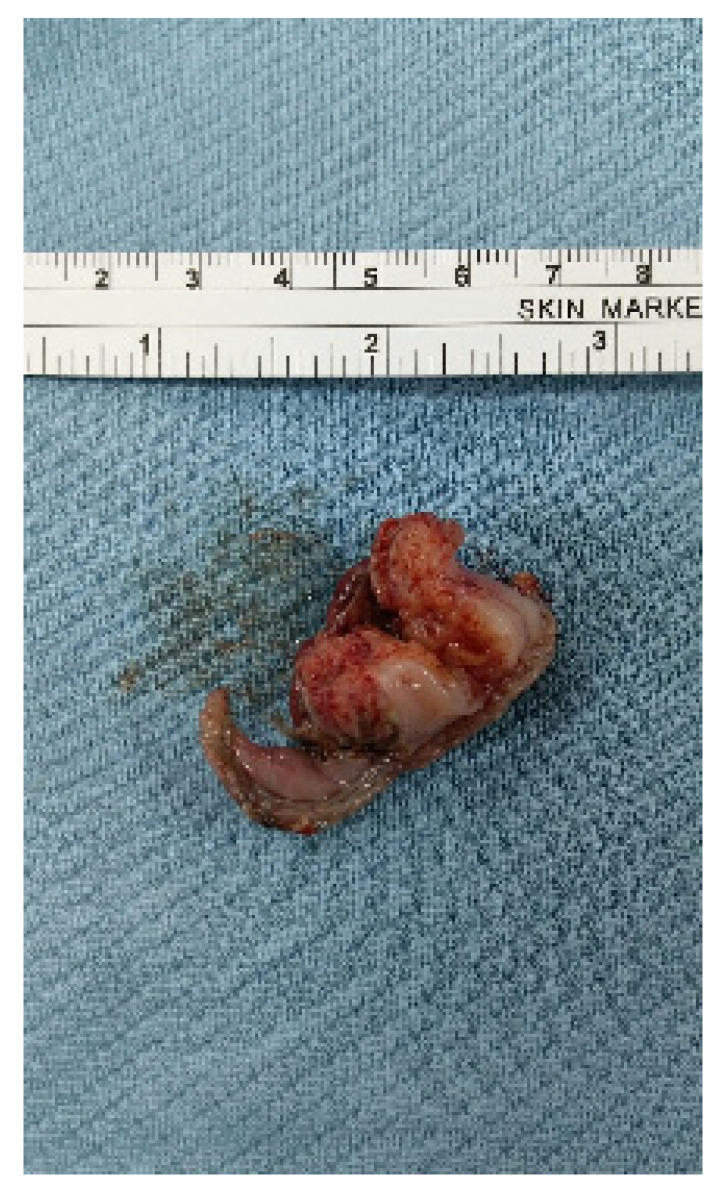
Gastric subepithelial tumor retrieved from specimen bag after laparoscopic endoscopic cooperative surgery.

**Figure 5 jcm-10-04423-f005:**
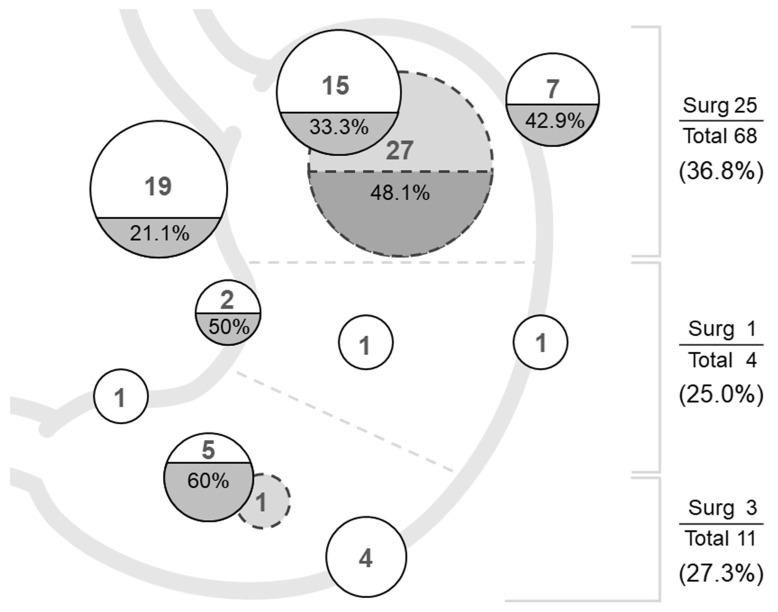
The location of the tumors and the percentage of the patients requiring surgery.

**Table 1 jcm-10-04423-t001:** Patient characteristics and peri-procedure parameters.

	ESD Only *n* = 56		Backup Surgery *n* = 27		*p* Value
Age, years	54.5	(48.0–63.0)	56.0	(50.5–63.3)	0.4222
Male gender	26	(46.4)	9	(33.3)	0.2606
Tumor location					0.8634
Upper	45	(80.4)	23	(85.2)	
Middle	3	(5.4)	1	(3.7)	
Low	8	(14.2)	3	(11.1)	
Tumor size					0.1455
≤2 cm	42	(75.0)	16	(59.3)	
>2 cm	14	(25.0)	11	(40.7)	
Layer of tumor depth					0.0792
Submucosa	6	(10.7)	0	(0.0)	
Muscularis propria	50	(89.3)	27	(100.0)	
Exophytic growth	0	(0.0)	10	(37.0)	<0.0001 *
Procedure time, mins	60.0	(40.0–90.0)	185.0	(152.0–236.8)	<0.0001 *
Length of stay, days	5	(4–6)	7	(7–8)	<0.0001 *
Clavien ≥ III complication	4	(7.1)	0	(0.0)	0.1571
Pathology					0.4387
Malignant/malignant potential	26	(46.4)	15	(55.6)	
Benign	30	(53.6)	12	(44.4)	

Values are median (interquartile range) or n (%); ESD: endoscopic submucosal dissection; * *p* < 0.05.

**Table 2 jcm-10-04423-t002:** Clinical data of five cases with delayed perforation after ESD for gastric subepithelial tumors.

Case	Age	Sex	Tumor Location	Size (cm)	Depth	ESD Time (minutes)	Time to Diagnosis (hours)	Symptom/Signs	Image Survey	Severity	Management	LOS (days)
1	56	F	U, les	1.0	SM	60	6	Fever	CXR: Bil. subphrenic air	Grade I	Conservative care	8
2	43	F	L, ant	2.6	SM	90	8	Localized abd. pain	CT: Pneumoperitoneum with few ascites	Grade I	Conservative care	8
3	69	F	U, ant	0.4	MP	33	47	Localized abd. pain	CXR: Bil subphrenic air	Grade III	Sono-guided aspiration	8
4	42	M	U, ant	0.5	MP	20	6	Hematemesis	CT: Pneumoperitoneum with internal bleeding	Grade IIIb	Laparoscopic gastrorrhaphy	8
5	52	M	U, post	0.5	MP	40	24	Fever with peritonitis	CT: pneumoperitoneum with few ascites	Grade IIIb	Laparoscopic gastrorrhaphy	9

F: female; M: male; U: upper third; L: lower third; Les: lesser curvature; Ant: anterior; Post: posterior; SM: submucosa; MP: Muscularis propia; ESD: endoscopic submucosal dissection; CXR: chest X-ray; CT: computed tomography; LOS: length of stay. Complication severity was graded according to Clavien–Dindo classification.

**Table 3 jcm-10-04423-t003:** Pathologic diagnoses of 83 patients with gastric subepithelial tumors.

	Total *n* = 83	ESD Only *n* = 56	Backup Surgery *n* = 27
Malignant or malignant potential			
GIST (%)	39 (47)	23	16
- Very low risk	22 (26.5)	16	7
- Low risk	12 (14.5)	3	8
- Intermediate risk	5 (6)	4	1
Neuroendocrine tumor	1 (1.2)	1	0
Hyperplastic polyp with focal high-grade dysplasia	1 (1.2)	1	0
Benign			
Leiomyoma (%)	31 (37.4)	25	6
Ectopic pancreas	3 (3.6)	1	2
Calcifying fibrous tumor	2 (2.4)	1	1
Lipoma	1 (1.2)	1	0
Inflammatory fibroid polyp	1 (1.2)	1	0
Gastritis cystica profunda	1 (1.2)	1	0
Plexiform fibromyxoma	1 (1.2)	0	1
Elastofibroma	1 (1.2)	1	0
Pyloric gland adenoma	1 (1.2)	0	1

Values are *n* (%).

## Data Availability

All data generated or analyzed during this study are included in this published article.

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
