# Peer review of "Endoscopic Resection for Gastric Subepithelial Tumor with Backup Laparoscopic Surgery: Description of a Single-Center Experience"

_jcm, 2021, doi:10.3390/jcm10194423_

Round 1

Reviewer 1 Report

General Comments:

In this manuscript, the authors retrospectively investigated the short and long-term outcomes of endoscopic resection and salvage surgery for gastric subepithelial tumor. The results suggested that surgical intervention was required in 27 of 83 cases (32.5%) and exophytic growth was a risk factor of conversion. The size, depth and location of tumor were also considered to be relative risk factors although they were not statistically significant. Furthermore, 5 of 56 patients who underwent only endoscopic resection had delayed perforation within 2 days. There were no recurrence and mortality in the follow-up period. In general, these findings have been reported in the previous studies and your study does not seem to include novel findings. Please find details below.

Specific recommendations for revision:

  1. The purpose of this study is unclear. What is the novelty in this study?
  2. In this study, the rate of conversion from endoscopic resection to salvage surgery seems too high. Was the indication of endoscopic resection appropriate? Moreover, delayed perforation was found in 5 of 56 patients who underwent only endoscopic resection. Did you not consider performing laparoscopic seromuscular reinforcement in those cases with high risks of delayed perforation despite performing endoscopic resection in the operation room?
  3. You mentioned that laparoscopic wedge resection was feasible in the failed cases with incomplete endoscopic resection in the conclusion section. However, I consider that converting to LECS can be a better option for minimizing resection and preserving function of the stomach because you have an endoscope and other equipment to perform ESD in the operation room. Why did you choose wedge resection with a stapler for salvage surgery?

Reviewer 2 Report

The authors conducted retrospective study to analyze patients who underwent endoscopic resection for gastric SETs with a high probability of surgical intervention and outline a feasible procedure in cooperation with salvage surgery.

These findings suggest an important data for future studies to evaluate minimally invasive resection of SETs.

However, I have some concerns that should be addressed regarding the study design, methodology, and interpretation of results.

Specific comments

a) Major

1) First of all, is "salvage surgery" a widely used or appropriate term? Since the patients in this study have already had a radical resection with ESD successfully, I'm more comfortable with the term "conversion surgery" rather than "salvage surgery", which implicates salvage surgery for cancer. If the term is already widely used, the authors should refer to the appropriate literature, and if it is newly defined in this paper, the authors should define it clearly.

2) This is a new surgical strategy using ESD with a back up of general anesthesia and surgeons for SETs for which there is currently no evidence to recommend it in the guidelines. The clinical benefit of this strategy cannot be demonstrated without a comparison of outcomes with guideline-recommend surgical resection. In the case of this study, a comparison with LECS will be necessary. The clinical results of LECS at the authors' institution or the studies of LECS already presented should be shown in the discussion. And in what ways is this surgery useful? In terms of frequency of complications, operative time, cost, or patient invasiveness?

3) The authors mentioned timely laparoscopic surgery can provide favorable outcomes. However, general anesthesia as a back up for complications is not a common concept. Gastric perforation, the major complication in this study, can be adequately treated by emergency surgery after the onset. Did the patient with the uncontrolled bleeding get into shock during ESD? It is important to ensure safety by general anesthesia and back up of surgeons, but for many patients, they will be over-treated by undergoing unnecessary general anesthesia. Moreover, timely laparoscopic surgery cannot prevent delayed complication. This risk-benefit balance of this surgery should be clearly explained.

4) This study shows high risk cases of complications with ESD. Shouldn't LECS be performed in these cases from the beginning? This point should also be discussed.

5) Line 58–61: Did this study consider all "consecutive" cases at the authors' institution between January 2013 and January 2021?

b) Minor

1) Line 356: This treatment is not recommended by the guidelines. Did the authors obtain sufficient informed consent from the patient before receiving this advanced treatment? Did the authors obtain an informed consent from the patients on providing clinical information for this paper?

2) The authors need to revise the text so that aim and conclusion clearly correspond to each other.

3) The authors need to revise the text abstract. In particular, what does two groups indicate, and what does salvage laparoscopic surgery indicate?

4) Check the references again, especially No20, and 23.

Round 2

Reviewer 1 Report

I consider that your revised manuscript still has some major issues. First, you cannot mention that endoscopic resection (ER) is a less invasive procedure than LECS because you compared ‘successful endoscopic resection’ with ‘failed endoscopic resection followed by laparoscopic wedge gastrectomy’, not LECS. You must compare ER or backup laparoscopic surgery with LECS performed in your institution during the study period. Second, I think that your strategy is not controlling a risk of gastric content spillage and peritoneal seeding of tumor cells in comparison with LECS because perforation occurred intraoperatively or postoperatively in many cases (31/83, 37.3%) in your study. Moreover, no data are indicated to prove your claim. In summary, your study is not designed appropriately and the results of your study are insufficient to support the conclusion.

Reviewer 2 Report

General comments

The authors responded point by point to Reviewer 2's comments. However, there are still some concerns.

Specific comments

1) The novelty of this study is not described clearly. What is the novelty in this study?

2) This study has no control cohort. The authors should show the data on complications and curative resection rates of only ESD for high-risk patients. Especially, the rate of intra-operative hemorrhagic shock was high in this study.

3) Why did the authors choose wedge resection as the treatment for gastric perforation and bleeding? Since the SMT has already been resected by ESD, simple suture closure will be the first choice to avoid gastric lumen stenosis.

4) The response for the reviewer 2's major comment 5 was not reflected in the text. This is an important point to confirm that the authors performed R0 resection by ESD for all cases.

5) Is the operator who performed the ESD the same in all cases? Is there any bias among different operators?

6) Line 261–264: There are several variants of LECS, and it is possible to avoid the risk of gastric content spillage and peritoneal seeding of tumor cells due to an intentional gastric perforation in small tumors of around 2 cm. In this study, the incidental perforation rate after endoscopic resection was high rate of 31.3% and the gastric content spillage and peritoneal seeding of tumor cells might happen in these cases. How do the authors explain this point?

7) Line 77–80: How did the authors determine the patient selection criteria?
